# Viscoelastic Properties of Polyelectrolyte Multilayers Swollen with Ionic Liquid Solutions

**DOI:** 10.3390/polym11081285

**Published:** 2019-08-01

**Authors:** Nagma Parveen, Pritam Kumar Jana, Monika Schönhoff

**Affiliations:** 1Institute of Physical Chemistry, University of Muenster, 48149 Münster, Germany; 2NRW Graduate School of Chemistry, University of Muenster, 48149 Münster, Germany; 3Interdisciplinary Center for Nonlinear Phenomena and Complex Systems, Université Libre de Bruxelles, 1050 Brussels, Belgium

**Keywords:** Kelvin-Voigt modeling, polyelectrolyte multilayers, ionic liquid, quartz crystal microbalance with dissipation, thickness, viscoelasticity, concentration regimes, swelling

## Abstract

Polyelectrolyte multilayers (PEM) obtained by layer-by-layer assembly can be doped with ionic liquid (IL) via the swelling of the films with IL solutions. In order to examine the mechanical properties of IL-containing PEM, we implement a Kelvin-Voigt model to obtain thickness, viscosity and elastic modulus from the frequency and dissipation shifts determined by a dissipative quartz crystal microbalance (QCM-D). We analyze the changes in the modeled thickness and viscoelasticity of PEI(PSS/PADMAC)_4_PSS and PEI(PSS/PAH)_4_PSS multilayers upon swelling by increasing the concentration of either 1-Ethyl-3-methylimidazolium chloride or 1-Hexyl-3-methylimidazolium chloride, which are water soluble ILs. The results show that the thickness of the multilayers changes monotonically up to a certain IL concentration, whereas the viscosity and elasticity change in a non-monotonic fashion with an increasing IL concentration. The changes in the modeled parameters can be divided into three concentration regimes of IL, a behavior specific to ILs (organic salts), which does not occur with swelling by simple inorganic salts such as NaCl. The existence of the regimes is attributed to a competition of the hydrophobic interactions of large hydrophobic ions, which enhance the layer stability at a low salt content, with the electrostatic screening, which dominates at a higher salt content and causes a film softening.

## 1. Introduction

Polyelectrolyte multilayers (PEMs) are thin films formed upon the adsorption of oppositely charged polyelectrolytes in a layer-by-layer (LbL) fashion [1]. Typically, PEMs are thin, yet mechanically robust. The thickness and mechanical strength of PEMs can be varied within a wide range, and typically vary between nm to several µm and between a few kPa to several MPa [2,3,4,5,6], respectively. This is possible by simply varying the number of deposited polyelectrolyte layers and/or intrinsic properties of polyelectrolytes, such as the charge density, molecular weight and structure [7,8,9,10,11,12,13]. Other factors can be employed to even induce post-preparative changes of the film properties: For example, the ionic strength and charge-to-size ratio of specific ions [14,15,16], pH stimulation [17,18,19], organic solvent [20] or inorganic/organic ions exposed to PEMs [21,22] have been shown to influence the thickness and mechanical strength of PEMs. Such tunable physical properties of PEMs make them suitable candidates for surface coatings [23], biological substrates [24,25] and electrochemical separators [26,27,28]. For the latter type of application, PEMs combined with a suitable electrolyte solvent can tremendously improve the transport of ions through the multilayer. An interesting class of such a solvent is given by ionic liquids (ILs). In the past decade, several studies have reported the potential of ILs in battery systems as they are a good solvent of conductive ions and typically have a large electrochemical window [29]. Earlier, we have shown that PEMs can be swelled with aqueous IL solutions, the swelling is controlled by the IL’s hydrophobicity [22], and the corresponding IL uptake can be controlled by stimulating a charge-excess in PEMs [18]. Similar to organic solvents [20] and inorganic ions [21], IL solutions induce swelling in PEMs, which is likely to influence their thickness and viscoelastic properties, i.e., the viscosity and elasticity. These physical properties describe whether a swollen multilayer is thin yet robust enough for a potential application as electrochemical separators. More importantly, changes in the multilayer thickness and viscosity/elasticity on the IL-induced swelling reflect the underlying molecular interactions, providing a better understanding of the microscopic processes governing swelling.

Experimentally, it is challenging to simultaneously extract viscoelastic parameters from in-situ swelling experiments, and often complementary techniques/methods, e.g., ellipsometry and nano-indentation are employed for such purposes [3,30]. In this work, we employed the Kelvin-Voigt model to PEMs swollen with an IL solution by using the corresponding frequency (Δ*f*) and dissipation (Δ*D*) shifts measured by a quartz crystal microbalance with dissipation monitoring (QCM-D). Because the quartz sensor used in QCM-D has piezoelectric properties, the acoustic wave propagated through a film adsorbed on the sensor is sensitive to the mechanical strength of the film [31]. The dependence of the Δ*f* and Δ*D* shift on the thickness and mechanical strength of a film adsorbed on a quartz sensor are well established [32], and varies slightly with the measurement conditions, for example, the presence of bulk liquid, and the mechanical property of the adsorbed film, e.g., rigid vs. viscoelastic [31,32]. For a viscoelastic film, a suitable model is the Kelvin-Voigt, which has previously been used to describe relatively rigid viscoelastic films, such as PEMs [2,7,33]. This modeling provides a platform to extract the film thickness, viscosity and elastic modulus (elasticity) from a single kinetic data set of QCM-D, allowing a reliable comparison of the parameters over different swelling cycles/steps of PEM. In recent years, multiple studies have implemented Kelvin-Voigt modeling to examine these parameters of PEMs during the layer assembly at different ionic strengths of the polyelectrolyte and/or washing solution [5,15,34] and upon the pH-induced swelling of the assembled layers [2,7,35]. However, the QCM-D-based Kelvin-Voigt modeling has not been implemented to characterize the post-preparative swelling of PEMs as a function of the concentration of salt solutions, particularly organic salt solution. Furthermore, the existing studies of PEMs have focused on the modeled thickness, and on either the viscosity or elastic modulus of the multilayers. In this study, we have analyzed all three Kelvin-Voigt parameters for PEMs swollen with aqueous solutions of an IL in a concentration-dependent manner. In particular, we used PEI(PSS/PDADMAC)_4_PSS and PEI(PSS/PAH)_4_PSS multilayer systems and two water-soluble ILs, i.e., 1-Ethyl-3-methylimidazolium chloride (EMIM) and 1-Hexyl-3-methylimidazolium chloride (HMIM). The PEM swelling was performed with increasing concentrations of aqueous IL solution in reversible cycles. We realized that the modelling requires that one takes into account the concentration-dependent viscosity of the IL solutions. The results show a monotonic increase of the multilayer thickness upon swelling with increasing IL concentrations, while the multilayer viscosity and elastic modulus show a non-monotonic change. Interestingly, these trends were observed in the case of swelling with IL and not with inorganic salt, i.e., NaCl. The changes in the modeled data can be divided into three concentration regimes of IL, which are controlled by the uptake of ions of IL and hydration water, modulating the hydrophobic interactions and electrostatic shielding in PEMs. 

## 2. Materials and Methods 

### 2.1. Polyelectrolytes

We used branched poly(ethylenimine) (PEI) (M_W_ ~25,000 g/mol, Sigma-Aldrich, Darmstadt, Germany), poly(diallyldimethylammonium chloride) (PDADMAC) (M_W_ ~ 100,000−200,000 g/mol, Sigma-Aldrich, Darmstadt, Germany), poly(allylamine hydrochloride) (PAH) (M_W_ ~ 120,000−200,000 g/mol, Polysciences, Hirschberg, Germany) and poly(sodium styrenesulfonate) (PSS) (M_W_ ~ 70,000 g/mol, Acros, Geel, Belgium). PSS was dialyzed and lyophilized, removing low molecular weight components, the other polymers were used as received. The polyelectrolytes were dissolved in a 0.25 M NaCl solution in ultrapure water, attaining a final polymer concentration (in terms of the monomer unit) of 10 mM. All solutions were adjusted to pH 6.5 by the addition of either 0.1 M NaOH or HCl (Honeywell Fluka, Bucharest, Romania). An aqueous solution of 0.25 M NaCl at pH 6.5 was used as a washing solution. All solutions were based on ultrapure water (ρ > 18 MΩcm) purified in a Milli-Q Academic purification stage (Merck Millipore, Darmstadt, Germany).

### 2.2. Ionic Liquids (ILs)

1-ethyl-3-methylimidazolium chloride (EMIM) (purity > 98%, dry) and 1-hexyl-3-methylimidazolium chloride (HMIM) (purity > 98.5%, dry) were purchased from Sigma-Aldrich (Darmstadt, Germany). They are water-soluble ILs, and were dissolved in ultrapure water at different molar concentrations for the swelling experiments.

### 2.3. Quartz Crystal Microbalance with Dissipation (QCM-D)

A quartz crystal microbalance with dissipation detection having four parallel flow chambers (E4, QSense, Göteborg, Sweden) was employed to monitor the assembly and swelling of PEMs on a gold-coated quartz crystal (fundamental frequency of 5 MHz, QSense). Prior to use, the crystals were cleaned by heating them in a cleaning solution for 20 min at 70 °C. The cleaning solution is a mixture of 25% ammonia solution (Acros), 35% hydrogen peroxide (VWR), and ultrapure water at a 1:1:5 volume ratio. QCM-D shifts were acquired only after establishing a constant baseline with ultrapure water at 20 ± 0.02 °C. Polyelectrolytes and IL solutions were injected to the quartz crystals at a flow rate of 200 and 50 µL/min, respectively. Five normalized overtones (3^rd^ to 11^th^) of the QCM-D responses, i.e., the frequency (Δ*f*) and dissipation (Δ*D*) shifts, were acquired.

### 2.4. Assembly of PEM

Polyanion and polycation solutions were alternatively injected to the quartz crystals through the flow cells, to form PEI(PSS/PAH)_4_PSS and PEI(PSS/PDADMAC)_4_PSS multilayers. A plateau in the Δ*f* and Δ*D* shifts of QCM-D indicates the accomplishment of a polyelectrolyte layer. The washing solution was injected after an adsorption time of ~10–15 mins to remove any excess salt and/or polymer. After the adsorption of 10 polyelectrolyte layers, the film was washed with ultrapure water at pH 6.5.

### 2.5. Swelling Cycles of PEM with IL Solutions

The swelling measurements were done by injecting IL solutions to PEMs adsorbed on the quartz crystal in QCM-D. Since the solution exchange in the QCM-D flow cell takes ~5 min at a flow rate of 50 μL/min, the IL solutions were flowed for 15 min, and then incubated at a zero flow rate until the Δ*f* and Δ*D* shifts reached a plateau. The plateau is defined by an absolute change in Δ*f* and Δ*D* by less than ±2 Hz and ±10^−6^, respectively. Typically, such a plateau was reached within 1 h from the injection of the IL solution. Otherwise, the observation was continued up to 8 h. Only in a few cases of swelling, no plateau was reached, even after this time. After each swelling step, ultrapure water was injected until a new plateau was reached, representing the deswelling of the multilayer. Such a swelling and deswelling step forms one swelling cycle. A series of swelling cycles with a successively increasing IL concentration was performed for each multilayer, employing EMIM, HMIM or NaCl solutions. All swelling cycles were repeated and averaged over at least three different multilayer samples. 

### 2.6. Kelvin-Voigt Modeling

While the Δ*f* shift of QCM-D in a first approximation represents the mass of an adsorbed film on a quartz crystal, the Δ*D* shift expresses the rigidity/softness of the adsorbed film. Although these relations are straightforward in the case of a rigid film in contact with a gas phase, yielding the Sauerbrey equation [36], they are somewhat complex for viscoelastic films in contact with a viscous bulk liquid, e.g., PEMs in the hydrated or swollen state. The dependence of QCM-D shifts on the thickness, viscosity and elasticity of hydrated or swollen PEMs can be computed by considering the bulk liquid effect and using a suitable model for multilayer viscoelasticity such as the Kelvin-Voigt model [32]. Equations (1), (2) represent the corresponding relation/dependence, where the viscosity and elastic modulus of the adsorbed film is formulated with the Kelvin-Voigt model:(1)$$\Delta f\approx -\frac{1}{2\pi \rho_{0}d_{0}}\left\{\frac{\eta_{2}}{\delta_{2}}+d_{1}\rho_{1}\omega -2d_{1}{\left(\frac{\eta_{2}}{\delta_{2}}\right)}^{2}\frac{\eta_{1}\omega^{2}}{\mu_{1}^{2}+\omega^{2}\eta_{1}^{2}}\right\}$$
(2)$$\Delta D\approx \frac{1}{2\pi f\rho_{0}d_{0}}\left\{\frac{\eta_{2}}{\delta_{2}}+2d_{1}{\left(\frac{\eta_{2}}{\delta_{2}}\right)}^{2}\frac{\mu_{1}\omega^{2}}{\mu_{1}^{2}+\omega^{2}\eta_{1}^{2}}\right\}$$
in the above equations, *f*, ρ, *μ*, *η* and *d* designate the resonance frequency (for the given overtone), density, elastic modulus, viscosity and thickness, respectively, and $\delta =\sqrt{\frac{2\eta }{\rho \omega }}$ (ω = 2π*f* is the angular frequency). The subscripts 0, 1, and 2 correspond to the quartz crystal, adsorbed film and immersion liquid (water or IL solution), respectively. 

We used the Kelvin-Voigt modeling feature of the Q-Tools software (Biolin Scientific, version 3.1.25.604, Sweden), which allows to compute the Δ*f* and Δ*D* shifts according to the above-mentioned equations and then to perform the least-square fitting of the measured QCM-D shifts. The density ($\rho_{2}$) and viscosity ($\eta_{2}$) of the immersion liquid, and the density of the PEM ($\rho_{1}$) were kept as fixed parameters in the modeling. The parameters to be determined, i.e., the thickness (*d*_1_), viscosity ($\eta_{1}$) and elastic modulus ($\mu_{1}$) of the PEM were kept free while performing the modeling. The parameters of the quartz crystal are constant and were taken into account in the Q-Tools software during the modeling. 

Concerning the PEM layer formation, the modeling was performed for the time covering the assembly of 10 layers, including the final washing step with water. Here, a dilute polymer solution or water is the bulk liquid in contact with the adsorbed film. The density and viscosity of pure water were therefore used as fixed parameters, and a well-reported mass density of the PEM (1070 kg m^−3^) [15,37] was used as the third fixed parameter (Table 1). 

For the swelling cycles in the IL solution, the modeling was only performed for data acquired during the exposure of the multilayer to the IL solution. Here, it was found that the viscosity of the IL solution, which is in contact with the film, has a major impact, since the Kelvin-Voigt modeling results of swollen multilayers were most sensitive to the viscosity of the bulk liquid, $\eta_{2}$, as observed from the corresponding least-square fit (Appendix A). Therefore, the viscosity of IL solutions was measured using a Rheometer (MCR 102, Anton Paar). The concentration-dependent values were used as the fixed parameter $\eta_{2}$ of the solution phase, while modeling PEMs swollen with the respective IL concentration. Other fixed parameters were the same as used in the case of modeling hydrated PEMs (Table 1). Using the fixed and free parameters, we performed the modeling over the 3^rd^ to 11^th^ QCM-D overtones as a global fit of all overtones with a joint set of parameters. The same approach was used for the modeling of PEMs swollen with NaCl solutions. 

## 3. Results

### 3.1. Bulk Liquid Viscosity 

At the concentrations of the IL and salt solutions employed for swelling, the salt affects the liquid viscosity rather strongly, which has to be taken into account in the Kelvin-Voigt model. Figure 1 shows the liquid viscosity data as obtained by rheological measurements. The viscosities increase linearly with the salt content. In addition, they scale with the cation hydrophobicity in the order HMIM > EMIM > Na. This indicates that it is hydrophobic interactions that cause the viscosity enhancement. Note that the anion (Cl^−^) is the same for all three salts. 

### 3.2. Kelvin-Voigt Modelling to Extract Thickness and Viscoelastic Parameters of PEM

The 7^th^ overtone of the frequency and dissipation shift is given in Figure 2a,b for the two layer systems investigated. In all cases, five overtones were fitted with the model, while for clarity we display here only one exemplary overtone. A full data set for a multilayer swelling at a given IL concentration including all fitted overtones is given in the supporting information (Appendix A). 

We first analyze the data during the layer formation, which is completed with the final flushing with pure water (see the arrow in Figure 2a,b). Up to this point, the QCM-D traces show a step-wise decrease in the frequency, Δ*f* (mass uptake, black line) and alternating changes in the dissipation, Δ*D* (damping, blue line), confirming the layer-by-layer assembly of the PEI(PSS/PDADMAC)_4_PSS and PEI(PSS/PAH)_4_PSS multilayers. Moreover, the frequency shifts extracted from the fits with the Kelvin-Voigt model (‘modeled’ QCM-D traces, red lines) are in agreement with the measured QCM-D shifts on the layer assembly. The same holds for the dissipation data. 

The thickness, viscosity and elastic modulus determined from the modeling show a stepwise increase on the assembly of subsequent layers (up to 600 and 350 min in Figure 2c,d, respectively). Upon the final washing step with pure water (see the position of the arrow), the thickness of the assembled layers decreases slightly along with a substantial increase in the film viscosity and elastic modulus. The removal of excess salt upon the final washing step with pure water is likely to be the reason behind the lowering of the film thickness and enhancement in the film viscoelasticity. According to our modeled data (Figure 2c,d, and Table 2), the assembled PSS/PDADMAC film is thicker and more viscoelastic in comparison to PSS/PAH, possibly because of the longer persistence length of PDADMAC (more rigid polymer). The thickness and elastic modulus of the PEM determined from our modeling are similar with those reported in the literature for similar multilayer systems. By employing techniques such as ellipsometry [38,39], or atomic force microscopy [40,41], multiple studies had shown that the thickness of PEMs assembled from about 10 layers of synthetic polyelectrolytes at an ionic strength of ~0.25 M NaCl, pH of 6.5 and relative humidity of 100% varies between ~25 to 50 nm. On the other hand, the viscosity and/or elasticity of hydrated PEMs were determined via an adhesion/force measurement using atomic force microscopy [3,5,42] or an acoustic impedance measurement using QCM [34] or upon performing the Kelvin-Voigt modeling of QCM-D data [2,5,15]. These studies report a viscosity of ~0.05 kg m^−1^ s^−1^ and an elastic modulus of ~2 to 5 MPa for multilayers assembled from about 10 adsorbed layers of polyelectrolytes. The agreement of our modeled data with these reported values in the literature validates the Kelvin-Voigt modeling performed in this study. 

### 3.3. Modeling of PEM Swollen with EMIM Solution

To the right-hand side of the arrow, Figure 2a,b show the measured QCM-D shifts of PEI(PSS/PDADMAC)_4_PSS and PEI(PSS/PAH)_4_PSS on the subsequent exposure to the EMIM solution and water. This reversible swelling cycle (IL solution/water) was performed with a stepwise increase of the EMIM concentration in subsequent swelling cycles, as indicated by the values in the graphs. A decrease in Δ*f* (mass uptake) and an increase in Δ*D* (damping) upon exposure to the EMIM solutions indicate an overall mass increase of the multilayers. The process appears to be reversible up to a certain EMIM concentration (see Appendix A and Ref. [22]). Figure 2a,b also display the agreement between the measured and modeled QCM-D shifts of the PEI(PSS/PDADMAC)_4_PSS and PEI(PSS/PAH)_4_PSS multilayers swollen with the EMIM solutions. Time-traces of the corresponding modeled thickness, viscosity and elastic modulus obtained are shown in Figure 2c,d. Here, the modeling was only performed within the duration at which the multilayer is in contact with an EMIM solution, resulting in the non-continuous lines in Figure 2a,b. To further analyze the salt dependence of the modeled parameters, we have extracted the thickness, viscosity and elastic modulus of the PEMs from these time-traces. The equilibrium values for each salt concentration were extracted at the time-point prior to the injection of water, resulting in an increase of Δ*f* and decrease of Δ*D* (see Figure 2a,b), and are given in Figure 3 in dependence of the EMIM concentration. 

### 3.4. Thickness of PEM upon Swelling with EMIM Solutions

The salt concentration-dependent equilibrium thickness of the films upon swelling increases with the EMIM concentration (Figure 3). For both PEI(PSS/PDADMAC)_4_PSS and PEI(PSS/PAH)_4_PSS multilayers, the thickness of the swollen film reaches up to ~1.5 times the film thickness of the non-swollen condition at zero EMIM concentration. In the case of PEI(PSS/PDADMAC)_4_PSS, the film thickness begins to decrease at ≥1.25 M EMIM concentration, indicating a partial dissolution of the multilayer (Figure 3a). Such a dissolution of PEI(PSS/PAH)_4_PSS multilayers was not observed (Figure 3b). These results match with the qualitatively discussed extent of swelling (change in the Δ*f* shift) of the multilayers and the stability limit concentration (c_stab_) of EMIM for the swollen multilayers reported in our earlier work [22]. The absolute change in the Δ*f* shift (mass uptake) and the modeled thickness at c_stab_ for the PSS/PADADMAC multilayers is relatively higher, indicating a larger uptake of IL in PSS/PADADMAC as compared to PSS/PAH. This is likely to be the reason behind the lower stability limit for the former multilayer.

### 3.5. Viscosity and Elastic Modulus of PEM upon Swelling with EMIM Solutions

Unlike the thickness, the viscosity and elastic modulus of the swollen multilayers show a non-monotonic change with an increasing EMIM concentration (Figure 4a,b). Their standard error is higher compared to that of the modeled thickness data. This probably arises from the coupled dependence of Δ*f* and Δ*D* on *µ_1_* and *η_1_*, while their dependence on *d*_1_ is simply linear (see Equations (1) and (2)), resulting in a relatively high sensitivity of these modeled parameters on the statistical deviation in the measured QCM-D shifts. Nevertheless, both the viscosity and elastic modulus of the PEI(PSS/PDADMAC)_4_PSS and PEI(PSS/PAH)_4_PSS multilayers increase by two- to four-fold at around 0.25 and 0.5 M EMIM concentration, respectively. Above this EMIM concentration, both the viscosity and elastic modulus drop to the initial viscoelasticity, i.e., of the hydrated multilayers. With a further increasing IL concentration, the film viscoelasticity either stays constant (PAH) or decreases further (PDADMAC). 

### 3.6. Viscoelastic Parameters for Different IL and Salt

Furthermore, we performed the modeling in cases of multilayer swelling with aqueous solutions of another IL, i.e., HMIM and also an inorganic salt, i.e., NaCl. Figure 5 presents the modeled thickness, viscosity and elastic modulus of swollen PEI(PSS/PDADMAC)_4_PSS vs. either the IL or NaCl concentration. The modeled data show fundamentally different trends between the cases of ILs and inorganic salt. In particular, for the IL solutions, the changes in the thickness, viscosity and elasticity of PEMs upon swelling can be divided into three concentration regimes of IL. In the first regime, which is between 0 to 0.5 M EMIM and 0 to 0.25 M HMIM, respectively, the thickness of the multilayer shows a small increase, whereas the film viscosity and elasticity increase by about four-fold. In the second regime, the multilayer stretches up to ~1.5 and ~2 times the hydrated film thickness, between 0.5 to 1.25 M EMIM and 0.25 to 0.7 M HMIM, respectively. In this concentration regime, both the film viscosity and elasticity either stay constant or slightly decrease (Figure 5a,b). In the third regime, which is above 1.25 M EMIM and 0.7 M HMIM, the film thickness shows a small drop, and the film viscosity and elasticity continue to clearly decrease below the initial values of the hydrated multilayer, indicating a partial dissolution of the multilayer (Figure 5a,b). In contrast, in the case of NaCl, the thickness of the swollen multilayers increases more or less linearly up to 1 M NaCl and then levels off (Figure 5c). The film viscosity stays more of less constant, and the elastic modulus decreases slightly with an increasing NaCl concentration. Overall, the changes in the film parameters of the multilayer are monotonous without showing distinct concentration regimes upon swelling with the NaCl solution. 

## 4. Discussion

Employing the Kelvin-Voigt modeling, we have investigated three physical parameters, i.e., the thickness, viscosity and elastic modulus of PEMs in the hydrated and swollen state. As described in the results section, the viscoelastic parameters can be varied over a wide range depending on the salt type and concentration. The general behavior pointed out in Figure 5 can be divided into three concentration regimes of IL (see the colored background). The thickness data in Figure 5a,b are consistent with our previous qualitative description based on the frequency shift data vs. IL concentration [22]. At low IL concentrations (first regime) the films slightly swell with an increasing IL concentration; then, in a second regime, the swelling effect becomes steeper, until at a limiting concentration (c_sw_) a final thickness is reached, which forms the transition to the third regime. The modeled thickness data show a similar behavior to the previous frequency shift data, because the thickness (d) and Δ*f* are strongly related (see Equation (1)). Although the frequency shift data represent only the fast kinetics, i.e., initial 5 mins from the IL injection [22], the corresponding shift contributes to >80% of the total frequency shift, particularly for the swelling cases (in the first and second regimes) of the PSS/PDADMAC multilayers.

It is interesting that the behavior of the viscosity and the elastic modulus also show distinct features according to these three regimes. Our results show that the PEMs become both more viscous and elastic (by two- to four-fold) in the first IL concentration regime. However, the values of these parameters drop to the initial values with small changes with an increasing IL concentration in the second regime and further in the third regime. Below, we will discuss these results in terms of a molecular interpretation. 

At the assembly conditions, the PEI(PSS/PDADMAC)_4_PSS and PEI(PSS/PAH)_4_PSS multilayers are charge-neutral and contain roughly 40% (v/v) water, i.e., in-situ water of PEMs [43]. Hydrated ions of IL or salt are driven into the multilayer by a difference in the chemical potential inside and in the solution, which is generated upon exposure to an IL solution. The influx of the IL solution in the PEMs reaches an equilibrium via the exchange of hydrated ions of IL and the in-situ water of the multilayer. Because of this exchange, the overall water content in the PEM shifts, i.e., it may either increase or decrease compared to the in-situ water content of the PEM. As a simple means to compare the softness and thus the water content of different films with different QCM-D frequency shifts, an analysis of the ratio of Δ*D* to Δ*f* was proposed [44,45]. For our present data, the Δ*D* to Δ*f* ratio vs. IL concentration is non-linear (Appendix A), indicating a non-linear increase in the water content of the PEM upon swelling with an increasing IL concentration. On the other hand, the uptake of the IL cation, as quantified in our earlier work [18], shows a linear dependence on the IL concentration. 

### 4.1. Viscoelastic Properties in First Regime

With the above information, we can argue that in the first concentration regime of the IL, the uptake of ions in the PEM is relatively low, and the overall water content in the PEM stays more or less the same as in the hydrated PEM (Scheme 1 of Figure 6). This leads to an overall small increase in the multilayer thickness. The large increase in the film viscosity and elastic modulus in this regime was observed in the cases of both ILs, but not for the case of NaCl (Figure 5c). This indicates that the mechanical properties of the PEM are sensitive to the uptake of IL solutions, or in other words they are specific to the interactions between the PEM and IL. Here, one plausible argument is that favorable hydrophobic interactions between the incorporated organic cation of the IL and polyelectrolytes of the PEM result in the increased film viscosity and elastic modulus [22]. In fact, since all three salts contain chloride as an anion, any differences in the film properties have to be attributed to the nature of the respective cation, based on the fact that the cations are indeed taken up by the film. In this regard, HMIM, which has a hexyl group, is more hydrophobic than EMIM, and a larger hydrophobicity effect is expected. This is evident in the fact that, while a similar increase in the multilayer thickness is observed at a comparable IL concentration (see Figure 5a,b), the viscoelastic parameters in the first regime are higher for HMIM compared to EMIM, if the same concentrations are compared (compare Figure 5a,b). 

### 4.2. Viscoelastic Properties in Second Regime

Above, we discussed that the uptake of the IL cation in the swollen PEM increases linearly with an increasing IL concentration spanning over the different IL concentration regimes. Accordingly, the hydrophobic interactions in the swollen PEM would be expected to increase over the IL concentration regimes, resulting in a continuous increase in the multilayer viscosity and elasticity. However, we observed an opposite trend, indicating that another interaction, i.e., most likely electrostatic interactions between the incorporated IL ions and the polyelectrolytes, plays a role here. The assembly strength of the swollen PEM may loosen with an increased uptake of small extrinsic IL ions due to the screening of the Coulomb attraction between the polyions, lowering the mechanical strength of the PEM. Small inorganic ions are known to exert such a screening effect. For example, Jaber et al. have shown a decrease in the multilayer elasticity upon their treatment with NaCl solutions [3], and we also observed a small decrease in the elastic modulus of the PEM upon swelling with the NaCl solutions (Figure 5c). Thus, we argue that the extent of hydrophobic and electrostatic interactions between the PEM and IL determines the observed changes in the thickness, viscosity and elastic modulus of the PEM swollen with a given IL concentration: Small amounts of hydrophobic ions contribute to the film stability by the additional hydrophobic interactions, enhancing the viscosity and shear modulus (see Figure 6, Scheme 1). At a larger ion content, however, the screening of the polyelectrolyte crosslinking becomes effective, yielding softer films with fewer new internal interactions (see Figure 6, Scheme 2). Consistent with this, the hydration water content in the PEMs is relatively high in the second regime, as indicated by the high Δ*D* to Δ*f* ratio (Appendix A) and the larger slope in the thickness vs. IL concentration (Figure 2). This increased water content may also contribute in lowering the film viscoelasticity of the swollen PEMs. 

### 4.3. Viscoelastic Properties in Third Regime

In the third IL concentration regime, the IL uptake reaches a level at which the charge pairs between two or more polyelectrolyte layers are further shielded, resulting in a partial dissolution of the swollen multilayer (Scheme 3 of Figure 6). Interestingly, the limiting IL concentration of the third regime, i.e., 1.25 M EMIM and 0.7 M HMIM for PEI(PSS/PDADMAC)_4_PSS extracted from the viscoelastic data matches with our earlier reported stability limit concentration (c_stab_) of the respective ILs extracted from the raw data of Δ*f*.

## 5. Conclusions

In conclusion, this article shows the application of the Kelvin-Voigt modeling to obtain the thickness and mechanical properties, i.e., viscosity and elastic modulus of PEMs swollen with an IL solution from their corresponding QCM-D shifts. The changes in the modeled parameters as a function of the IL concentration can be distinguished in three concentration regimes of ILs. In the first regime, the multilayers become more viscous and also more elastic, along with a minimal increase in their thickness. In the second regime, the PEMs’ thicknesses increase steeply but their viscosity and elastic modulus drop to the initial values. In third regime, the PEMs’ thicknesses almost level off, and their viscosity and elastic modulus decrease further. Our data show that changes in the modeled parameter of the PEMs in accordance to these regimes are specific to ILs, which are organic salts. We argued that, while hydrophobic interactions between the IL and PEM contribute to the increased viscosity/elasticity in the first regime, the electrostatic shielding effect upon a large uptake of IL ions competes with the hydrophobic interactions in the second regime, screening the charge pairs between polyelectrolytes in the PEM. The latter essentially affects/modulates the assembly strength between polyelectrolytes in PEMs, resulting in an increase of the thickness and a drop in the viscosity and elasticity of the multilayers. These competing effects are specific to large hydrophobic ions and do not occur upon swelling with a simple salt such as NaCl.

In addition to an intrinsic interest, the determination of PEM thickness and viscosity and/or elasticity, and finding the correlation between them at a swollen condition is crucial for their application in various fields. For the application as an electrochemical separator, PEMs doped with ILs or conductive ions must be thin yet mechanically robust enough. Our results present the correlation between the thickness and viscosity/elasticity of PEMs doped with ILs of different concentrations, which may provide the basis to obtain a good balance between the thickness and mechanical strength of the swollen films for their potential applications. 

## Figures and Tables

**Figure 1 polymers-11-01285-f001:**
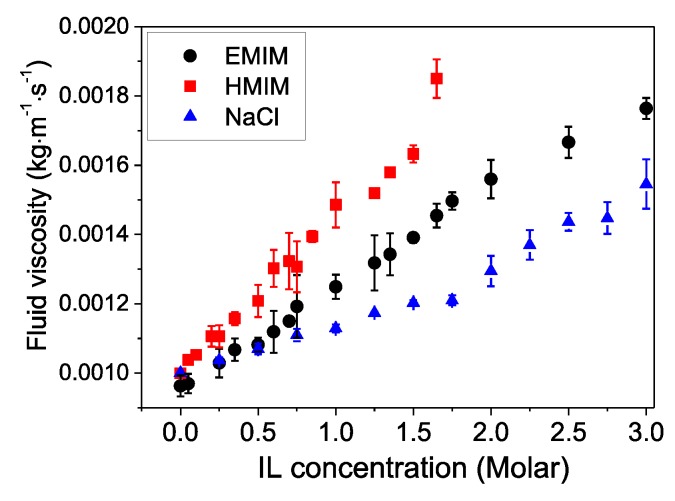
The viscosity of the aqueous solutions of ILs, i.e., EMIM and HMIM, and NaCl, as a function of their concentration.

**Figure 2 polymers-11-01285-f002:**
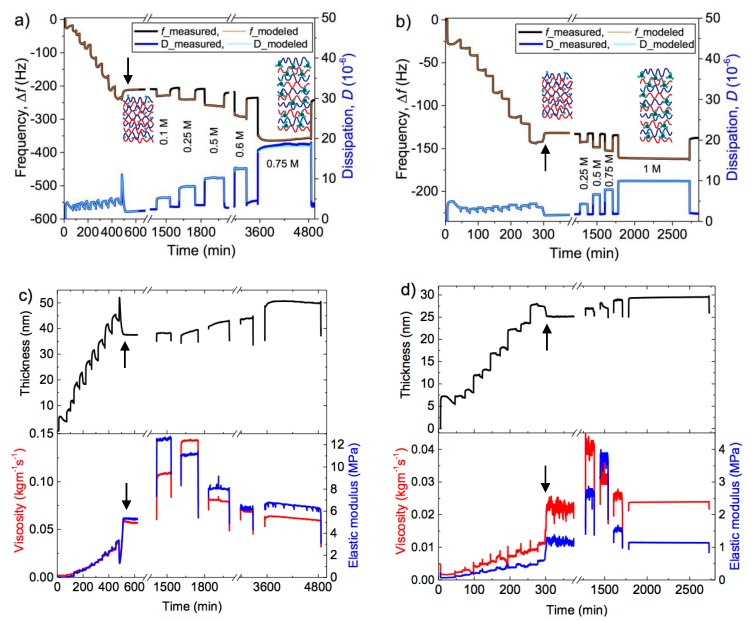
Measured and modeled QCM-D frequency and dissipation shift (7th overtone) upon assembly of (**a**) PEI(PSS/PDADMAC)_4_PSS and (**b**) PEI(PSS/PAH)_4_PSS and their subsequent swelling with the EMIM solution at the EMIM concentrations indicated. Modeled thickness, viscosity and elastic modulus upon layer assembly and subsequent swelling of (**c**) PEI(PSS/PDADMAC)_4_PSS and (**d**) PEI(PSS/PAH)_4_PSS multilayers. The arrows indicate the end of the assembly, where flushing with pure water was performed, followed by the salt swelling cycles. Note that only one representative overtone is displayed, while 5 different overtones were employed in a global fit.

**Figure 3 polymers-11-01285-f003:**
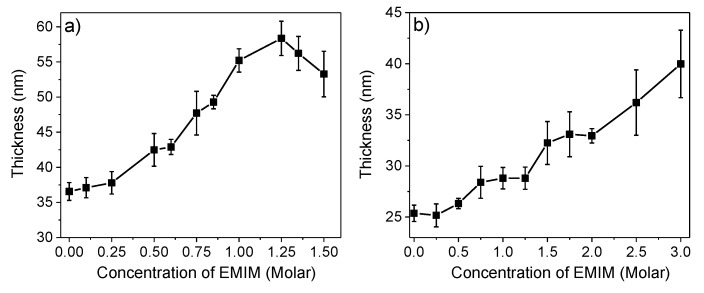
Modeled thickness of the (**a**) PEI(PSS/PDADMAC)_4_PSS and (**b**) PEI(PSS/PAH)_4_PSS multilayers vs. the concentration of EMIM solution used for the multilayer swelling in reversible cycles. The error bar is the standard error obtained from averaging over 3 data sets at the respective EMIM concentration.

**Figure 4 polymers-11-01285-f004:**
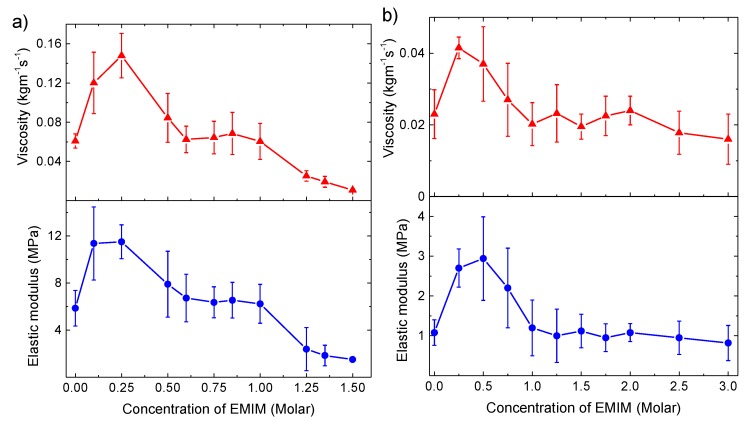
Modeled viscosity and elastic modulus of the (**a**) PEI(PSS/PDADMAC)_4_PSS and (**b**) PEI(PSS/PAH)_4_PSS multilayers vs. the concentration of EMIM solution used for the multilayer swelling in reversible cycles. The error bar is the standard error and calculated over 3 modeling data at the respective EMIM concentration.

**Figure 5 polymers-11-01285-f005:**
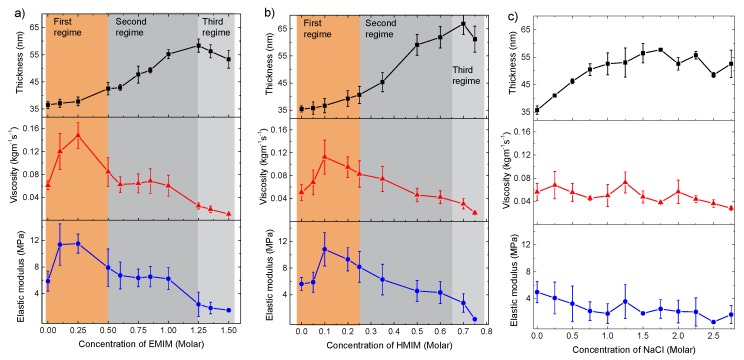
Modeled thickness, viscosity and elastic modulus of PEI(PSS/PDADMAC)_4_PSS upon swelling with the aqueous solution of (**a**) EMIM (**b**) HMIM and (**c**) NaCl at the given concentrations. Changes in the modeled parameters in different concentration regimes of ILs are highlighted. The error bar is the standard error from averaging over 3 data sets at the respective salt concentration.

**Figure 6 polymers-11-01285-f006:**
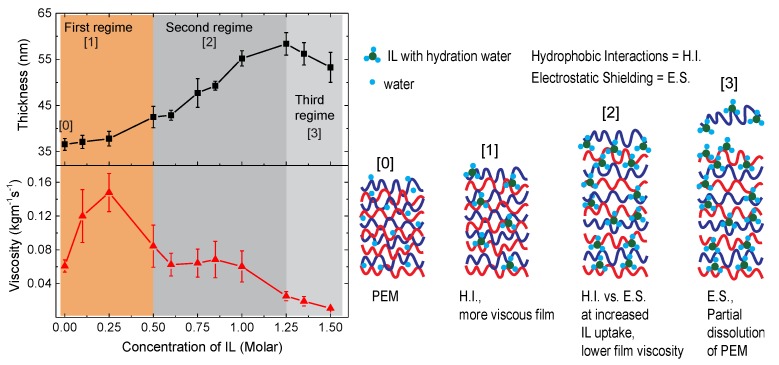
Modeled thickness and viscosity of a multilayer upon swelling with aqueous solutions of an IL. Different concentration regimes of the IL are highlighted, and schemes (0 to 3) illustrate the effect of hydrophobic interactions and electrostatic shielding on the film thickness, viscosity and/or elasticity of a multilayer swollen with IL solutions over the different concentration regimes.

**Table 1 polymers-11-01285-t001:** Fixed parameters and range of free parameters used for the Kelvin-Voigt modeling of PEI(PSS/PADMAC)_4_PSS and PEI(PSS/PAH)_4_PSS multilayers either in hydrated state or upon exposure to an IL solution.

Fixed Parameters of Modeling		
Parameter	Hydrated Polyelectrolyte Multilayers (PEM)	PEM Swollen with IL Solution
Density of bulk liquid, $\;\rho_{2}$	1000 kg m^−3^	1000 kg m^−3^
Viscosity of bulk liquid, $\;\eta_{2}$	0.001 kg m^−1^ s^−1^	Viscosity of IL solution at a given IL concentration, see Figure 1
Density of PEM, $\;\rho_{1}$	1070 kg m^−3^	1070 kg m^−3^
**Range of the Free Parameters of Modeling**		
PEM Thickness, *d_1_*	0.1 to 1000 nm	
PEM Viscosity, $\eta_{1}$	0.005 to 1 kg m^−1^ s^−1^	
PEM Elastic modulus, $\mu_{1}$	0.005 to 100 MPa	

**Table 2 polymers-11-01285-t002:** The average thickness, viscosity and elastic modulus of hydrated PEI(PSS/PADMAC)_4_PSS and PEI(PSS/PAH)_4_PSS multilayers obtained from the Kelvin-Voigt modeling.

PEM	Thickness, *d*_1_ (nm)	Viscosity, *η*_1_ (kg m^−1^ s^−1^)	Elastic Modulus, *μ*_1_ (MPa)
PEI(PSS/PDADMAC)_4_PSS	36.7 ± 0.9	0.06 ± 0.01	4.8 ± 1.8
PEI(PSS/PAH)_4_PSS	25.3 ± 0.5	0.02 ± 0.01	1 ± 0.3

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
