# Peer review of "Viscoelastic Properties of Polyelectrolyte Multilayers Swollen with Ionic Liquid Solutions"

_polymers, 2019, doi:10.3390/polym11081285_

Round 1

Reviewer 1 Report

In this paper, authors prepared polyelectrolyte multilayer thin films (PEM), PEI(PSS/PADMAC)4PSS and PEI(PSS/PAH)4PSS and then PEM was doped with ionic liquid (IL) via the swelling.  The authors studied the mechanical properties of IL-containing PEM such as thickness, viscosity and elastic modulus by a QCM.  This is an interesting paper.  I have found some questions which are desired to be answered before the publication.

(1)  The authors used two ionic liquids, 1-Ethyl-3methylimidazolium chloride (EMIM) and 1-Hexyl-3methylimidazolium chloride (HMIM).  The authors told in p10 L 360-362 that small amounts of hydrophobic ions contribute to film stability by the additional hydrophobic interactions, enhancing viscosity and shear modulus, see Figure 6, Scheme 1.  HMIM, which has hexyl group, is expected to be more hydrophobic than EMIM.  However, in First regime - Figure 5a,b, HMIM is less effective in increasing the viscosity and the elastic modulus than that of EMIM. 

Authors should discuss in more detail about differences of HMIM and EMIM in more detail.

(2)  The authors told in p 5 L 204-206 that the assembled PSS/PDADMAC film is thicker and more viscoelastic in comparison to PSS/PAH, indicating a higher stress resistance of the former film.  The authors also told in p 7 L 269-270 that the film viscoelasticity either stays constant (PAH) or decreases further (PDADMAC).
  Authors should discuss in more detail about differences of PDADMAC and PAH in more detail.  Similar discussion also should be given in “3.4. Thickness of PEM upon swelling with EMIM solutions”, whereas you referred your earlier work.

Minor point: in p10 L338-341

The large increase in the film viscosity and elastic modulus in this regime was observed in cases of both ILs, but not for the case of NaCl.  This indicates that the mechanical properties of PEM are sensitive to the uptake of IL solutions, or in other words, specific to the interactions between PEM and IL (Figure 5c).
  The position of (Figure 5c) could be as follows:

The large increase in the film viscosity and elastic modulus in this regime was observed in cases of both ILs, but not for the case of NaCl (Figure 5c).  This indicates that the mechanical properties of PEM are sensitive to the uptake of IL solutions, or in other words, specific to the interactions between PEM and IL.

Author Response

In this paper, authors prepared polyelectrolyte multilayer thin films (PEM), PEI(PSS/PADMAC)4PSS and PEI(PSS/PAH)4PSS and then PEM was doped with ionic liquid (IL) via the swelling.  The authors studied the mechanical properties of IL-containing PEM such as thickness, viscosity and elastic modulus by a QCM.  This is an interesting paper.  I have found some questions which are desired to be answered before the publication.

We thank the reviewer for the comments.

(1)  The authors used two ionic liquids, 1-Ethyl-3methylimidazolium chloride (EMIM) and 1-Hexyl-3methylimidazolium chloride (HMIM).  The authors told in p10 L 360-362 that small amounts of hydrophobic ions contribute to film stability by the additional hydrophobic interactions, enhancing viscosity and shear modulus, see Figure 6, Scheme 1.  HMIM, which has hexyl group, is expected to be more hydrophobic than EMIM.  However, in First regime - Figure 5a,b, HMIM is less effective in increasing the viscosity and the elastic modulus than that of EMIM.  Authors should discuss in more detail about differences of HMIM and EMIM in more detail.

On first sight it might seem as if HMIM is less effective in enhancing viscoelastic parameters, if the values at the corresponding maxima are compared. Within error, the values of the maxima are, however, the same. If the parameters are compared at the same IL concentration, however (note the different axis scaling in Fig. 5 a and b), it is evident that HMIM has a larger effect, as expected.

We have clarified the comparison made here by adding some sentences in line 364-368 discussing the difference in hydrophobicity of ILs and the corresponding effect in the viscoelastic properties of PEM in the first IL concentration regime.

(2)  The authors told in p 5 L 204-206 that the assembled PSS/PDADMAC film is thicker and more viscoelastic in comparison to PSS/PAH, indicating a higher stress resistance of the former film.  The authors also told in p 7 L 269-270 that the film viscoelasticity either stays constant (PAH) or decreases further (PDADMAC).
  Authors should discuss in more detail about differences of PDADMAC and PAH in more detail.  Similar discussion also should be given in “3.4. Thickness of PEM upon swelling with EMIM solutions”, whereas you referred your earlier work.

We have rephrased the sentence in line 212-214 and stated the plausible reason behind the difference in PSS/PAH and PSS/PDADMAC multilayers. Further, in Section 3.4 we have added sentences in 269-272 discussing the difference in the multilayer stability.

Minor point: in p10 L338-341

The large increase in the film viscosity and elastic modulus in this regime was observed in cases of both ILs, but not for the case of NaCl.  This indicates that the mechanical properties of PEM are sensitive to the uptake of IL solutions, or in other words, specific to the interactions between PEM and IL (Figure 5c).
  The position of (Figure 5c) could be as follows:The large increase in the film viscosity and elastic modulus in this regime was observed in cases of both ILs, but not for the case of NaCl (Figure 5c).  This indicates that the mechanical properties of PEM are sensitive to the uptake of IL solutions, or in other words, specific to the interactions between PEM and IL.

It is changed as indicated by the Reviewer.

Reviewer 2 Report

The work by Parveen et al. presents an interesting study about the effect of ionic liquid in the viscoelastic properties of polyelectrolyte multilayer. The work is well performed and the results are physically sound. However, authors ignore completely previous effort in the characterization of the viscoelastic properties of polyelectrolyte multilayers using the QCM-D. Authors should introduce before publication the following references in their reference list:

-Soft Matter 5 (2009) 2130-2142

-Journal Physical Chemistry C 116 (2012) 15474-15483

-Physical Chemistry Chemica Physics 13 (2011) 18200-18207

-Langmuir 26 (2010) 11494-11502

-Beilstein Journal of Nanotechnology 7 (2016) 197-208

After addition such references, the work will be ready for publication.

Author Response

The work by Parveen et al. presents an interesting study about the effect of ionic liquid in the viscoelastic properties of polyelectrolyte multilayer. The work is well performed and the results are physically sound. However, authors ignore completely previous effort in the characterization of the viscoelastic properties of polyelectrolyte multilayers using the QCM-D. Authors should introduce before publication the following references in their reference list:

-Soft Matter 5 (2009) 2130-2142

-Journal Physical Chemistry C 116 (2012) 15474-15483

-Physical Chemistry Chemica Physics 13 (2011) 18200-18207

-Langmuir 26 (2010) 11494-11502

-Beilstein Journal of Nanotechnology 7 (2016) 197-208

After addition such references, the work will be ready for publication.

According to the suggestion of Reviewer2, the first four references are added in line 37 and 39, respectively. The fifth reference (BJN) is of little relevance , as it does not contain any viscoelastic data.
